# The extratropical tropopause - Trace gas perspective on tropopause definition choice

Sophie Bauchinger<sup>1</sup>, Andreas Engel<sup>1</sup>, Markus Jesswein<sup>1</sup>, Timo Keber<sup>1</sup>, Harald Bönisch<sup>2</sup>, Florian Obersteiner<sup>2</sup>, Andreas Zahn<sup>2</sup>, Nicolas Emig<sup>3</sup>, Peter Hoor<sup>3</sup>, Hans-Christoph Lachnitt<sup>3</sup>, Franziska Weyland<sup>3</sup>, Linda Ort<sup>4</sup>, and Tanja J. Schuck<sup>1</sup>

**Correspondence:** Sophie Bauchinger (bauchinger@iau.uni-frankfurt.de)

#### Abstract.

Aircraft measurement campaigns such as IAGOS-CARIBIC and HALO missions are invaluable sources of trace gas observations in the extratropical Upper Troposphere and Lower Stratosphere (exUTLS), providing simultaneous measurements of multiple substances. To contextualise these observations, the use of dynamic coordinate systems such as tropopause-relative coordinates is highly beneficial. Different approaches to define the tropopause are commonly used in studies, based on either differences in chemical composition, dynamical parameters, or temperature gradients between the troposphere and stratosphere. We examine how different tropopause definitions influence the climatology and seasonality of trace gas observations. Meteorological parameters used in this analysis are obtained from ERA5 reanalysis data interpolated to the flight tracks. Our findings indicate that the thermal tropopause results in larger composition variability near the tropopause but shows the sharpest overall transition. Potential-vorticity thresholds of 1.5 or 2.0 PVU result in vertically displaced distributions. The 3.5 PVU threshold best represents the transport barrier as indicated by low variability and strong curvature around the tropopause. Tracer-based tropopauses using O<sub>3</sub> or N<sub>2</sub>O can be used effectively to differentiate between the troposphere and stratosphere without the use of additional model data. A chemical tropopause tied to a mid-latitude ozone climatology was shown to return a meaningful tropopause-relative coordinate. An investigation of individual flights showed that the tropopauses calculated from model data did not represent small-scale structures well, while the 'in-situ' chemical tropopauses provided more meaningful results. For the calculations of an N<sub>2</sub>O-based statistical tropopause, however, the case studies highlighted the importance of carefully setting initial parameters.

#### 1 Introduction

The chemical and radiative properties of the extra-tropical Upper Troposphere and Lower Stratosphere (ExUTLS) are heavily influenced by atmospheric transport and mixing processes. However, in-situ observations in this complex region are rare, leading to significant uncertainties in our understanding of processes such as stratosphere-troposphere exchange (STE) or long-term

<sup>&</sup>lt;sup>1</sup>Institute for Atmospheric and Environmental Sciences, Goethe University of Frankfurt, Frankfurt, Germany

<sup>&</sup>lt;sup>2</sup>Karlsruhe Institute of Technology, Institute of Meteorology and Climate Research, Karlsruhe, Germany

<sup>&</sup>lt;sup>3</sup>Johannes Gutenberg University of Mainz, Institute for Atmospheric Physics, Mainz, Germany

<sup>&</sup>lt;sup>4</sup>Atmospheric Chemistry Department, Max Planck Institute for Chemistry, Mainz, Germany

trends in atmospheric circulations. Despite growing scientific interest and research in this region over the past few decades, key knowledge gaps remain: For example, while changes in the strength of the jet streams and the Brewer-Dobson circulation have been observed, the extent and implications are not yet well understood. Ozone trends in the exUTLS are also difficult to quantify due to the complex interplay between processes such as large-scale circulations and STE, which contributes to a large natural variability. These uncertainties extend to pollutant transport and consequently the overall impact of aerosols or short-lived trace gas emissions. Reducing these uncertainties is crucial for improving our ability to model current and future climate change. To address these, long-term analysis of trace gas distributions is essential for identifying trends, conducting STE studies, and assessing the strength of transport barriers such as the tropopause.

30

40

Analysis of long-term phenomena in trace gas distributions depends to some part on our ability to identify dynamically equivalent air masses over long periods of time. As the atmosphere is a highly dynamic system, the location of transport barriers such as the tropopause and the subtropical and polar jet are not fixed. When analysing long-term data sets, comparing measurements using geometric coordinates (altitude, latitude, longitude) thus introduces additional variability. Improvements can be achieved through the use of coordinates representing natural flow pathways in the atmosphere such as potential temperature, especially in combination with equivalent latitude in the stratosphere (Pan et al., 2012). In the UTLS region, however, changes in trace gas gradients at the tropopause can have a significant impact on the comparability of measurements.

Studies of the UTLS therefore benefit from using tropopause-relative coordinates, which can again be expressed in terms of pressure, potential temperature or other parameters. Millán et al. (2024) systematically show the effects of tropopause-relative vertical coordinates for ozone data in a variety of data sets. Coordinates relative to the subtropical jet were also tested, but this approach did not result in a significant improvement in grouping air masses. On the other hand, the study highlighted the benefits obtained by using potential temperature over pressure for UTLS studies of ozone.

Although tropopause-relative coordinates are widely used in UTLS studies, there is no consensus on the "best" coordinate system. The variety of available tropopause definitions and vertical coordinate options makes it challenging to select the most appropriate system for specific questions, especially considering that the advantages and disadvantages of different tropopause definitions depend on the particular research context. The tropopause has historically been defined by the World Meteorological Organisation (WMO) using the temperature lapse rate (WMO, 1957), later definitions based on atmospheric dynamics (Reed, 1955; Tinney et al., 2022; Turhal et al., 2024) or chemical composition (Bethan et al., 1996; Sprung and Zahn, 2010; Prather et al., 2011; Assonov et al., 2013) emerged.

This work will provide an overview of the most commonly used tropopause definitions in exUTLS studies. To allow comparability with previous studies, we put a focus on tropopause definition parameters as available in published data sets, rather than separate implementations. Measurements of well-mixed and long-lived trace gases in the upper troposphere and lower stratosphere can be used to quantify how well any tropopause definition represents the differences between the two atmo-

spheric layers. We show the differences in homogeneity around the different tropopauses by calculating binned variability and calculating the curvature at the tropopause. For a holistic discussion of this topic, we look at two perspectives: First, we compare tropopauses on a climatological scale and then look at individual events. For the climatological analysis, we focus on data collected on-board a passenger aircraft during the second phase of the IAGOS-CARIBIC (Commercial Aircraft for Regular Investigation of the atmosphere Based on an Instrument Container) project. We use the temporal resolution of the flask samples for all CARIBIC analyses, as measurements of N<sub>2</sub>O are not available at higher resolution. We then take a case-study approach to test the climatological results using individual flights of the PHILEAS campaign of the German research aircraft HALO. The trace gas observations are combined with meteorological parameters from ERA5 reanalysis data interpolated onto the flight tracks.

# 2 Data

During the second phase of the IAGOS-CARIBIC project, 19 instruments were fitted into a strongly modified LD11 airfreight container, which was loaded onto a passenger aircraft. The dataset used for this study includes 278 individual flights, taking place regularly between 2005 and 2020. Through a specially designed inlet system, incoming air was distributed to a variety of in-situ measurement devices and a flask sampler (Brenninkmeijer et al., 2007; Petzold et al., 2015). Owing to the instruments onboard the aircraft, simultaneous high resolution ( $< 10 \, s$ ) measurements of  $O_3$ , CO and many other substances were taken throughout the entire project. Flask sample data was collected using the air sampling device TRAC (Triggered Retrospective Air Collector) (Brenninkmeijer et al., 2007; Schuck et al., 2009), which consists of 14 glass cylinders. In 2010, flask sample collection was expanded with 88 stainless-steel cylinders with the device HIRES (HIgh REsolution Sampler). Samples were taken every 30 to 60 minutes by pressurising the containers to 4.5 bar using metal bellow pumps upon reaching a pressure level of less than 480 mbar. Total sample collection times were 0.5– $1.5 \, min$ , resulting in a spatial resolution of 7–21 km. After collection, the following constituents of the air samples were measured at laboratories across Europe: greenhouse gases, nonmethane hydrocarbons, halocarbons and isotopic composition. The  $N_2O$  measurements used here are part of the greenhouse gas data set and were carried out with an HP 6890 gas chromatograph coupled to an electron capture detector. (Schuck et al., 2009, 2012). As part of the whole-air-sampler (WAS) data set, the  $N_2O$  data has been made available in Schuck and Obersteiner (2024).

Meteorological parameters for each observation were obtained by interpolating ERA5 reanalysis data onto the flight tracks and time stamps. The reanalysis data used has a temporal resolution of 6 hours, horizontal grid spacing of 1° and 137 vertical levels, which approximately corresponds to a spacing of 500 m (Hersbach et al., 2020). The available variables include potential temperature, potential vorticity, as well as various parameters describing the distance between the flight track and multiple definitions of the tropopause. The distance between flight track and thermal tropopause was derived by matching the model's pressure levels to the measured in flight pressure, while dynamic tropopauses, PV and equivalent latitude were first interpolated onto isentropes using potential temperature and then interpolated onto the flight track. The time resolution of the model data

was chosen to match the 10 s intervals of the merged high resolution CARIBIC data set. Further interpolation of the model data, as well as high resolution trace gas measurements, was carried out to match the WAS data timestamps. For consistency, all data from the CARIBIC project used in this study are on WAS time resolution.

Measurements taken as part of the CARIBIC project make up a unique data set due to the regularity of flights, the large geographical coverage of the northern hemisphere and the simultaneous observations of a multitude of substances and parameters. Particularly of interest for this study is that in mid- and high latitudes, the cruise altitude of commercial aircraft roughly coincides with the location of the tropopause / UTLS region. We therefore limit the data set to latitudes > 30°N, with the added benefit that all tropopause definitions in this comparison are well-defined in this region. For this subset, we find that at least 65% of measurement points are located within 2 km of the tropopause for all definitions.




For the case studies in Section 4.4, we use observations from the PHILEAS campaign in August and September 2023 aboard the research aircraft HALO (Riese et al., 2024). The transfer flights between Germany and Alaska were chosen as they provide significant geographical coverage and the routes taken were not tailored to explore special features in the atmosphere. A large benefit of this data set is the availability of high resolution in-situ trace gas measurements, including  $N_2O$ . The measurements of  $N_2O$  were carried out using the University of Mainz Airborne Quantum Cascade Laser Spectrometer (UMAQS) with a total uncertainty of 0.13 ppb after calibration and comparison to NOAA working standards (Müller, 2015). Measurement data from the PHILEAS campaign is available on the HALO database. As for the CARIBIC data, ERA5 reanalysis data including meteorological parameters and tropopause information were interpolated onto the PHILEAS flight tracks.

Quantification of the homogeneity of the data set due to dynamically induced variability can best be done by using a substance with clear characteristic of its atmospheric distribution and profile. In this study, we use focus on O<sub>3</sub> due to the well-understood and distinct atmospheric profile coupled with the ability to measure this substance with good precision at high resolution. O<sub>3</sub> data of CARIBIC and PHILEAS were obtained using a custom-built UV photometer with a typical uncertainty of 2% (or 2 ppb absolute) and a time series resolution of 0.25 Hz (Zahn et al., 2012; Obersteiner, 2024). The O<sub>3</sub> data from CARIBIC used here is part of the merged data set on 10 s time resolution (Zahn et al., 2024), while the measurements during PHILEAS are available upon request via the HALO Database (HALO, 2025).

# 3 Tropopause definitions

Multiple tropopause definitions fell in and out of favour over time with technological advancements and improved observation coverage (Gettelman et al., 2011). Table 1 shows an overview of the three main types of tropopause definitions that are in active use by the meteorological community, the advantages and disadvantages of each being the focus of this paper. Traditionally, the tropopause has been calculated using the temperature gradient obtained from atmospheric temperature profiles (WMO, 1957). Later, the ability to model dynamic processes on large scales allowed the formulation of dynamic tropopause definitions

**Table 1.** Overview of commonly used tropopause definitions and their basis.

| Definition                                    | Based on:                                                               | Data from:           |  |
|-----------------------------------------------|-------------------------------------------------------------------------|----------------------|--|
| Thermal / WMO                                 | Temperature lapse rate                                                  | Observations / Model |  |
| Dynamic                                       | PV surface (e.g. 1.5, 2.0 or 3.5 PVU)                                   | Model                |  |
| Chemical: O <sub>3</sub> and N <sub>2</sub> O | Climatology, tracer-tracer correlation, statistical baseline evaluation | Observations         |  |

based on PV. Early proposals of a dynamic tropopause reference the isentropic gradient of PV, however for simplicity, dynamic tropopauses were largely adopted as constant PV surfaces. The PV gradient tropopause has gained more attention again through work of Kunz et al. (2011) and more recently by Turhal et al. (2024). Another approach discussed in Tinney et al. (2022) uses the gradient of potential temperature to define a tropopause similar to the temperature gradient, which was shown to identify sharper gradients in O<sub>3</sub> where these definitions do not coincide. For aircraft campaigns, the calculation of the distance to the local thermal and dynamic tropopauses relies on the availability of model data. This is due to vertical temperature profiles or potential vorticity curtains not being available in the data measured onboard.


With increasing availability of trace gas profiles through sondes and aircraft campaigns, the definition of chemical tropopauses became possible. The tropopause acting as a transport barrier, especially outside the tropics, results in steep gradients of some tracers in this region. Over the years, tracers such as  $O_3$  (Bethan et al., 1996; Sprung and Zahn, 2010),  $N_2O$  (Assonov et al., 2013; Umezawa et al., 2014), or modelled tracers, for example the e90 tracer with an e-fold decay time of 90 days, (Prather et al., 2011) have been used in literature with varying implementations. In concept, these chemical tropopauses are better able to identify small-scale structures and mixing events than modelled tropopauses which first have to be interpolated from lower temporal and spatial resolution model data.

For this work, thermal and dynamic tropopauses based on meteorological dynamic variables are calculated from ERA5 reanalysis data. The tropopause height and distance to the flight track are calculated for multiple vertical coordinates, such as pressure, geopotential height or potential temperature. Two separate approaches to define a chemical tropopause based on trace gas measurements are also presented.

# 3.1 Thermal tropopause

- In the troposphere, the temperature generally decreases with increasing height, as less energy absorbed by the surface reaches higher air masses. This relationship, however, is inverted in the stratosphere. The definition of a *thermal* troposphere was officially formulated by the WMO (1957) with the following conditions:
  - 1. Identify the lowest level at which the temperature lapse rate is below  $2 \,\mathrm{K \, km^{-1}}$ .

- 2. The condition must be fulfilled that the average lapse rate between that level and all other higher levels within two kilometres does not exceed  $2 \, \mathrm{K \, km^{-1}}$ .
- 3. Secondary or tertiary tropopauses may be identified if these conditions are met again at higher altitudes.

# 3.2 Potential vorticity-based tropopause






The PV value of an air parcel describes its capacity to rotate, thus giving information on the combination of dynamics and stratification of the atmosphere. Similarly to potential temperature, PV is conserved in frictionless and adiabatic flow. PV is defined as the dot product of stratification and absolute vorticity and its unit is the potential vorticity unit (PVU) defined as  $\frac{10^{-6}\,\mathrm{K}\,\mathrm{m}^2}{\mathrm{kg}\,\mathrm{s}}\equiv 1\,\mathrm{PVU}.$  Due to stratification above the tropopause, PV experiences a strong increase from troposphere to stratosphere in the extra-tropics, allowing for the definition of a dynamic tropopause. While early proposals of PV-based tropopause definitions were focussed on the PV gradient, most studies chose to present the dynamic tropopause using PV surfaces to simplify evaluating the tropopause. In 1986, the WMO identified 1.5 PVU as the ideal cut-off for the troposphere, although later studies found higher PV values to more accurately represent the tropopause. Throughout history, a variety of PV values have been used, with 1.5, 2.0 and 3.5 PVU standing out as the most commonly used thresholds; for a discussion see e.g. Hoerling et al. (1991). Furthermore, Kunz et al. (2011) found that the ideal PV threshold values may be different for the two hemispheres. The value of the PV threshold is correlated with the altitude of the tropopause, so that of these values, 1.5 PVU represents the lowest tropopause and therefore the strictest tropospheric criterion. The threshold values are positive in the northern hemisphere and negative in the southern hemisphere, as absolute vorticity changes sign. The dynamic tropopause is not defined near the equator, as (potential) vorticity experiences a singularity in that region. Consequently, the dynamic tropopause is often represented by a fixed value of potential temperature in the tropics, commonly set to 380 K, or alternatively combination with the thermal tropopause have been suggested (Wilcox et al., 2012). For this analysis, we focus only on extratropical observations north of 30°N.

### 3.3 Chemical tropopauses

As some longer-lived trace gases have a strong gradient across the tropopause, this characteristics of their vertical profile may be used to define a *chemical* tropopause. There are approaches using many different gas species and combinations thereof. This analysis will focus on two definitions: 1) using a mid-latitudinal climatology of  $O_3$  to give the distance to the thermal tropopause and 2) a statistical baseline filter based on  $N_2O$  observations. Average seasonal profiles for  $O_3$  and  $O_3$  in CARIBIC data can be seen in Appendix A. By using chemical tropopauses over reanalysis data, the same data set may be used both for the definition of a tropopause as for the trace gas or parameter of interest. The mismatch between model resolution as well as the potential for modelled features to be slightly offset in location to observations can therefore be avoided. However, the use of chemical tropopauses is naturally limited by the availability of high-quality measurements.

### 3.3.1 Ozone-based tropopause






A typical vertical profile of  $O_3$  can be characterised by a fairly well-mixed troposphere, a sharp gradient across the tropopause and increasing mixing ratios with height in the lower stratosphere. The profile is controlled by the mixing of descending  $O_3$ -rich stratospheric air and tropospheric air poor in  $O_3$ , leading to a consistent and strong increase within the LMS (Fischer et al., 2000). Making use of the relative abundance in ozone sonde data,  $O_3$  observations can be used to find a distance between a climatological tropopause and the measurement point.

We use the chemical $O_3$  tropopause as described in Sprung and Zahn (2010). Firstly, a climatology of  $O_3$  data is created using sonde profiles from Hohenpeissenberg, linking  $O_3$  mixing ratio to a position above the local thermal tropopause. This tropopause definition can then be applied to observational data by selecting the corresponding climatology per month. A measure of the distance to the tropopause is then obtained by finding the bracketing climatological values and interpolating the chemical $O_3$  tropopause to the measured  $O_3$  value. Note here that the relative altitude is not given for values below 60 ppb or results lower than 1.5 km below the tropopause, as the chemical $O_3$  tropopause does not yield meaningful results for these values. It should additionally be noted that the representativeness of this single-station ozone climatology is not explored in detail here. The sensitivity to latitudinal differences in the ozone distribution may be of interest for future work.

# 3.3.2 Nitrous Oxide-based tropopause

The greenhouse gas  $N_2O$  is emitted primarily by soil and oceans, whereas the main loss mechanism is photolysis in the stratosphere. In combination with the trace gas' life-time of 123 (104–152) years (SPARC, 2013), this results in the atmospheric profile of  $N_2O$  to have a well-mixed background in the troposphere, a strong cross-tropopause gradient in mid- and higher latitudes, and decreasing mixing ratios with increasing altitude in the stratosphere (Bisht et al., 2021). This characteristic profile allows the definition of a chemical tropopause based on  $N_2O$  measurements, where lower  $N_2O$  measurements are indicative of stratospheric air masses. Defining a chemical $N_2O$  tropopause using trends measured at Mauna Loa, Umezawa et al. (2014) found this definition to be approximately equivalent to a dynamical tropopause with a 2.5 PVU threshold. Defining a tropopause based on  $N_2O$  observations has the benefit of being able to use the same observational data set as other substances at the focus of further study.

For this study, we separate stratospheric data from the tropospheric observations with higher mixing ratios using the statistical baseline filtering approach from Schuck et al. (2018). In an initial pre-flagging step, the iterative filter identifies and removes data points more than 3% lower than the respective tropospheric value by referencing background station observations. A tropospheric baseline is then fitted to the remaining points. Any measurements with mixing ratios further than 2 standard deviations below this fit are identified as stratospheric in origin. This process repeats until the limit criterion is reached, which we define as a change of less than 10% of the standard deviation of the baseline data compared to the previous iteration. By

modifying this parameter, the chemical<sub>N2O</sub> tropopause can be adjusted for a specific data set.







Similarly to a  $O_3$ -based tropopause, the chemical $_{N_2O}$  definition does not show sensitivity below the tropopause due to the well-mixed nature of these tracers in the troposphere. As the statistical baseline filter does not contain information on the vertical position of measurement, this tropopause can moreover not be expressed in traditionally used vertical coordinates such as height, pressure or potential temperature. This additionally hinders the study of parameters such as trends in the location of the tropopause or LMS mass. However, the difference between the  $N_2O$  value to its corresponding baseline can be treated as an approximation for vertical displacement from the tropopause for some applications.

As for other tracer-based tropopause definitions, the chemical $_{N_2O}$  tropopause depends on how well a substance can be measured and the availability of the data. For this statistical filter to work properly, a significant fraction of the data needs to be tropospheric for the evaluation of a baseline. This could potentially be mitigated by careful combination of multiple data sets.

# 3.4 Key differences in tropopause definition methodologies

Different tropopause definitions capture different aspects of atmospheric structure and composition, depending on the underlying data sources and calculation methods. Chemical tropopauses are based on the current local composition of the air. Apart from changes to chemical composition, specific measurements themselves are therefore agnostic to large-scale meteorological structures or dynamics. The time-resolution of the measurements then controls at what scale processes can be resolved - while flask sample data gives the average composition of the air over the sampling time, in-flight measurements with high temporal resolution also resolve small-scale structures more clearly. In contrast, dynamic and thermal tropopauses for aircraft data rely on reanalysis or model output and reflect large-scale atmospheric structures. The spatial and temporal coherence can be an integral part of transport studies, however makes them less sensitive to small-scale features.

Owing to limitations in temporal and spatial resolution, global reanalysis data does not provide detailed information on small-scale local phenomena. Features such as localised tropospheric or stratospheric intrusions, mixing events or narrow streamers may be missed by modelled tropopauses, but leave a signature in the chemical composition. Joint analysis of the measurements of tracers like  $O_3$  or  $N_2O$  and other substances measured on the same platform can then provide more detailed information than combining the measurements with a separate reanalysis dataset. Measurement-based definitions may therefore provide more accurate vertical attribution in certain cases. However, dynamic barriers such as the jet streams and PV contours are not directly represented.

Taking stratosphere-troposphere-exchange (STE) as an example, one can easily see the differences between the approaches. When stratospheric and tropospheric air masses mix during STE events, air with intermediate chemical characteristics is produced. Chemical tropopauses then assign a corresponding intermediate height to this air mass, which directly represents this

mixed characteristic. On the other hand, dynamic or thermal tropopauses may treat STE as a process happening below or above a static barrier, if temperature gradient or PV around the tropopause are not directly affected. These differences in positioning of mixed air masses relative to the tropopause are important to keep in mind, for example for attribution studies.

Tropopause folds are another example of how differently complex structures may be represented using different definitions. Definitions for secondary tropopauses exist for dynamic and thermal tropopauses, which may identify tropopause folds. These are dependent both on the sharpness of the edge of the fold and the resolution of the data used to identify these. For chemical definitions, however, the identification of an air masses stratospheric or tropospheric characteristic is the central focus. Here again it is important to understand that chemical definitions return *representative* positions relative to the tropopause.

For this work, we focus on comparing the performance of tropopause definitions on the homogenisation of measurements using tropopause-relative coordinates. Clear separation of air masses with tropospheric or stratospheric characteristics then reduces the variability within these atmospheric layers. In the following sections, the impact of different tropopause definitions on the homogeneity of ozone measurements will explored in more detail.

### 4 Results




### 4.1 CARIBIC data characterisation

To illustrate the differences between tropopause definitions, we first characterise the CARIBIC data set and its suitability for UTLS studies. In Fig. 1 the average seasonal distance of data points to the respective tropopause ( $\Delta z_{TP}$ ) is shown. As the geopotential height of the flights does not vary considerably for all seasons and latitudes, the seasonal and latitudinal differences in  $\Delta z_{TP}$  are due to variations in the location of the tropopause. For all tropopauses, the sampling at lower latitudes can be characterised as more tropospheric. Overall, the distribution of measurement points shows that the CARIBIC data set is suitable for extratropical upper troposphere and lower stratosphere (exUTLS) studies due to significant sampling in the region around the tropopause in all seasons. The relationship between tropopause height and measurement points both in seasonality and latitudinal dependence varies between tropopause definitions. Here we use the geopotential height relative to the tropopause as vertical coordinate to compare the thermal, the three dynamic and the chemical<sub>O3</sub> tropopauses.

The chemical<sub>N2O</sub> tropopause is based on a statistical baseline filter based on the observed mixing ratio, which does not result in a clearly defined location of the tropopause. Instead, the difference between measurements and the baseline evaluated for that point in coordinates (the 'residual') is shown in Fig. 1 to give an approximate comparative value. In this plot, a negative difference indicates data points with lower N<sub>2</sub>O than the baseline, which corresponds to a larger fraction of stratospheric air. One can see that neither the seasonal average nor its standard deviation go significantly below zero, which shows the absence of downward sensitivity for this tropopause definition. The chemical<sub>N2O</sub> tropopause data shown in Fig. 1 therefore mainly shows

Figure 1. Latitude-binned (5°) mean distance of whole-air-sampler measurements to the respective tropopause across the whole CARIBIC measurement period. Panels (a) and (b)–(f) show the distance in geopotential height, while in panel (b) the difference between  $N_2O$  observations and the statistically evaluated baseline of the chemical $N_2O$  is given. The grey shading indicates the standard deviation of the seasonal average.

the seasonality of stratospheric measurements in the CARIBIC data set, while the tropospheric data has a smaller impact on these values. Notably, the chemical<sub>N2O</sub> presents very differently to other tropopause definitions, with the displacements in summer being close to the average, while the thermal and dynamic definitions then result in lower  $\Delta z_{TP}$  than other seasons.

Figure 1 also shows that the distribution of measurement points at higher latitudes is in agreement for the dynamic<sub>3.5</sub> and the thermal tropopause. In this region, the seasonality of the thermal and dynamic definitions follows the same overall structure, where flight paths are closest to the lower lying tropopause in summer. The difference between distributions using dynamic tropopauses with different thresholds is evident mostly through a vertical displacement, while the seasonality and latitudinal

trends are very similar. Towards low latitudes, the thermal tropopause results in larger seasonal differences than the dynamics definitions, with the average distance between measurements and the tropopause ranging from less than one to almost four kilometres below the tropopause. In this region, the influence of the subtropical jet (STJ) is expected to lead to dynamic changes which may not be well represented by the temperature gradient.

In contrast, both chemical tropopause definitions result in lower seasonal variability at latitudes between 30–50°N. This is likely in large part due to sampling bias, as the flight path height relative to the tropopause is not meaningful for ozone values below 60 ppb. In this comparison, the chemical tropopauses therefore exhibit a lower variability at lower latitudes, as the remaining data set is more homogenous regarding mixing ratios. Other tropopause definitions result in higher variability in this region, as the calculation of average distance between flight track and tropopause is independent on the ozone mixing ratio and therefore less sensitive to this filtering bias. In the CARIBIC data set, this effect is weakest at higher latitudes, where the flight path is on average higher above the tropopause.

At high latitudes, the chemical $_{O_3}$  tropopause tends to be closer to the flight track in winter and further away in spring, while the summer and autumn values stay close to the average displacement. We believe the difference in seasonality between chemical $_{O_3}$  and thermal/dynamic tropopauses to be an effect of using a mid-latitudinal ozone climatology to calculate the chemical $_{O_3}$  tropopause. Using ozone observations to find the displacement to the tropopause in the polar regions through comparison to a mid-latitudinal climatology is likely to result in an underestimation of the displacement to the tropopause in high latitudes in winter. The different sensitivity of tropopauses to downwelling may additionally impact the seasonal variations. Changes in the chemical composition through increased downwelling in some seasons will immediately be represented by the chemical tropopauses, while changes in PV appear with more complexity.

As a consequence of the high number of data points around the tropopause and varying tropopause heights, the total number of data points in the troposphere or stratosphere varies significantly between tropopause definitions. Depending on the definition, the total share of tropospheric data points ranges from 29 % to 48 %. Figure 2 shows that the ratios are statistically in good agreement for all seasons between the thermal, dynamic<sub>3.5</sub> and chemical<sub>O3</sub> tropopauses. There are some slight differences between these definitions. In winter, the dynamic<sub>3.5</sub> tropopause identifies a larger fraction of tropospheric air in the data set, while in spring the stratospheric fraction is higher for the chemical<sub>O3</sub> tropopause than the other two. However, the overall seasonal variations are very similar for these definitions. Note here that the chemical<sub>O3</sub> tropopause is not completely independent of the thermal tropopause: As outlined in Sect. 3.3.1, the climatology of the chemical<sub>O3</sub> definition is calculated as a distance to the thermal tropopause. While the differences between tropopauses seasonalities are small, they may be an indication of different sensitivities of specific definitions to seasonal changes in the atmosphere, for example diabatic transport processes. Following from the displacement of the vertical tropopause location in the atmosphere, the tropospheric ratios obtained for different dynamic tropopauses decrease with lower PVU-thresholds. Of each data point identified as tropospheric

**Figure 2.** Tropospheric fraction per season as identified by different tropopause definitions in the CARIBIC data set 2005–2020 on whole-air-sampler time resolution. A fraction of 1 indicates an even split of tropospheric and stratospheric measurements.

by the dynamic<sub>3.5</sub> tropopause, only 70 % are also below the dynamic<sub>2.0</sub> tropopause and 61 % below the dynamic<sub>1.5</sub> tropopause.

It is expected that with careful choice of the corresponding threshold values, the dynamic tropopause should coincide with trace gas-based definitions. This is because trace gas distributions roughly follow surfaces of adiabatic frictionless flow, which the dynamic tropopause represents. As the meteorological variables are calculated as part of a global model, however, small features may either not be resolved or appear slightly offset from the measured atmospheric dynamics. Moreover, Fig. 2 shows that while the two tracer-based definitions resulted in similar seasonalities of  $\Delta z_{TP}$  over latitude, the seasonality of tropospheric fractions presents quite differently for the two chemical tropopauses.

While the chemical $_{0_3}$  tropopause results in a similar tropospheric fraction as the thermal and dynamic $_{3.5}$  tropopauses, the tropospheric fraction is much higher in autumn for the chemical $_{N_2O}$  tropopause. The ratios obtained using the chemical $_{N_2O}$  tropopause exhibit a much stronger seasonality than other tropopause definitions, with only 30.1% of data points in spring identified as tropospheric, compared to 60.5% in autumn. This increased tropospheric fraction may be caused by the sensitivity of the statistical baseline algorithm on the underlying dataset. The weakening of the subtropical jet and corresponding flushing of tropical air into the extra-tropics results in higher  $N_2O$  mixing ratios in the UTLS. The baseline algorithm may then identify the mixed air masses as tropospheric, while other definitions are less sensitive to these chemical composition changes. This effect appears to then not be apparent for the chemical $_{O_3}$  tropopause, as the climatology at the tropopause already includes this seasonality. Another effect causing this difference could be the seasonal variation of  $N_2O$  emissions propagating upwards. A systematic investigation of the statistical baseline algorithm across different seasons and conditions is recommended to better

335

340

understand this behaviour.

To check for biases in the WAS data set, we repeat this analysis with the high resolution CARIBIC data on 10 s timestamps as well as for the 40–60°N latitude subset. We find that these results are in agreement with the low-resolution analysis
both in overall tropospheric ratios and seasonal variations. A comparison between potential temperature differences between
flight track and dynamic/thermal definitions results in the same overall trends for seasonality and latitudinal dependence. In
this coordinate system, however, the differences between seasons are not dependent on the latitude to the same extent as with
geopotential height. We continue using WAS timestamps and geopotential height for the following analyses to allow the use of
both the N<sub>2</sub>O measurements and the height-based chemical<sub>O3</sub> tropopause data.

# 4.2 Homogeneity of vertical distributions

We now use ozone observations to investigate how well trace gas distributions across the atmosphere are characterised using different tropopause definitions. Figure 3 shows interpolated CARIBIC ozone data sorted into troposphere and stratosphere. Note that over 4% of the points sorted into the troposphere by the thermal tropopause have ozone mixing ratios over 200 ppb, compared to 1.5% for the dynamic<sub>3.5</sub> tropopause and less than 0.1% for all other definitions. The influence of higher ozone measurements sorted into the troposphere leads to a higher variability of potential temperature bins, which are here indicated through horizontal bars. In contrast, lower PV-thresholds of the dynamic tropopause result in a reduction in the number of tropospheric points, which leads to lower mean O<sub>3</sub> and variability per potential temperature bin. The dynamic<sub>1.5</sub> and dynamic<sub>2.0</sub> tropopauses however also result in sorting a significant number of observations with low ozone mixing ratio into the stratosphere. For these definitions, 16% and 12% of stratospheric ozone mixing ratios were below 100 ppb, compared to 6% or less for other tropopause definitions. This results in strong decreases of the mean mixing ratio across all seasons and increases in the variability in regions close to the tropopause. Overall, the mean ozone value is shown to strongly depend on the tropopause definition, leading to a range of 60 ppb in the stratosphere and 21 ppb in the troposphere.

365

360

355

To evaluate the effectiveness of grouping homogeneous air masses by different tropopause definitions, we explore the variability of vertical profile bins in more detail. Similar to before, ozone measurements are identified as either tropospheric or stratospheric, however now they are binned on a grid of geopotential height (0.5 km) and latitude (1° N). For each bin containing more than 5 data points, the variability of those observations is calculated. To evaluate which tropopause definitions results in better homogenisation of the ozone data, i.e. an overall lower binned variability, we show the frequency distribution of variabilities in Fig. 4. The distributions can roughly be quantified by applying a log-normal fit. The fitted curves as well as the mode of the resulting fit are indicated by the black lines. In an idealised case, the distribution would be dominated by the natural variability of ozone with a minimisation of the variability introduced by changes in tropopause height. In the troposphere, we expect lower variability values for definitions that are more effective in filtering out stratospheric data points.

375


**Figure 3.** Ozone mixing ratios over potential temperature sorted into tropospheric and stratospheric air using different tropopause definitions. The average measurement values and standard deviation are shown for each potential temperature (5 K) interval.

The results of this log-normal fit of variability distributions for all seasons and tropopause definitions is shown in Fig. 5. While the overall number of tropospheric data points is similar for the thermal, dynamic<sub>3.5</sub> and chemical<sub> $O_3$ </sub> tropopauses, clear differences exist in the binned variability. For all seasons, the spread of tropospheric bin variabilities of ozone is largest with the thermal tropopause with corresponding high modes in spring and summer, although in autumn and winter the chemical<sub> $N_2O$ </sub> tropopause results in a higher variability mode. The chemical<sub> $N_2O$ </sub> tropopause results in relatively broad distributions of variability in autumn and winter, which likely follows from the higher tropospheric fraction in these seasons than other tropopause definitions.


The influence of chemical processes and much higher mixing ratios leads to a larger natural variability of ozone in the stratosphere. The stratospheric variability distributions are therefore much less sensitive to differences between tropopause

**Figure 4.** Distribution of tropospheric and stratospheric variability of CARIBIC ozone data. Measurements are separated using different tropopause definitions and binned onto a grid of latitude (1° N) and geopotential height (0.5 km). The distribution of variability of the resulting bins is shown as a histogram with log-normal fit applied to it. The mode of the distribution is indicated by a dashed line leading to the peak of the fitted curve.

definitions. Figure 5 however clearly shows the reduced variability of ozone in autumn and higher variability in spring and summer. For an alternative representation, the variabilities of binned ozone data are also shown in coordinates relative to the tropopause in Appendix B.

# 4.3 Vertical profiles and curvature at the tropopause

The previous sections have shown that there are significant differences between tropopause definitions even in coordinates such as potential temperature or geopotential heights. To further explore these differences, we now investigate the impact when using tropopause-relative coordinates. Figure 6 shows that the cross-tropopause gradient of  $O_3$  is strongest in spring and summer and weakest in autumn and winter for all tropopause definitions with a clearly defined vertical coordinate.

The chemical $_{N_2O}$  tropopause in Fig. 6b is shown using the difference between  $N_2O$  mixing ratio to the statistically evaluated baseline for each measurement. While this coordinate gives some information on the vertical position of the measurement point, this representation does not show the seasonality of the ozone vertical profile in the way other tropopause definitions

Figure 5. Mode and 68/95% interval widths of log-normal fit on distribution of variability in 2D-binned  $O_3$  measurements on WAS resolution. Binning was carried out for each season in latitude (1°) x geopotential height (0.5 km). The distribution is shown for (a) the troposphere and (b) the stratosphere as identified by different tropopause definitions. The large dots indicate the mode of the variability distribution while the shading shows the 68% and 95% intervals of the log-normal fit.

do, but rather the correlation between  $O_3$  and  $N_2O$ . One can see that the correlation does not have a significant seasonality, however the variability of  $O_3$  vs.  $\Delta N_2O$  strongly increases in winter.


All other tropopause definitions show clear seasonal variations in mean mixing ratios in troposphere and stratosphere. The overall ozone mixing ratio at a given height above the tropopause is lowest in autumn and winter and highest in summer and spring, with a qualitatively similar seasonality in the troposphere. The seasonal differences tend to become more pronounced the deeper in the stratosphere the measurements are being taken. While there is a clear separation between seasons for the

thermal and dynamic definitions, the winter and summer profiles resulting from the chemical $O_3$  tropopause are similar in value. Overall, the seasonal vertical profiles are alike for the dynamic and thermal tropopauses.

The variability in the tropospheric bins tends to be smaller for all definitions than in the stratosphere. As before, this is due to overall much higher mixing ratios in the stratosphere as well as chemical processes introducing natural variability. For the chemical $O_3$  tropopause, however, the variability in spring and autumn are much smaller than other tropopause definitions higher above the tropopause. The variability is highly impacted by the use of ozone both as the basis for calculating the tropopause as well as substance at the focus for this analysis and can therefore not be directly compared.




The curvature at the tropopause of trace gases with distinct tropospheric and stratospheric characteristics can be used as a measure of how well a given definition characterises the transport barrier. A stronger curvature indicates a sharper transition between troposphere and stratosphere, reflecting a clearer separation between tropospheric and stratospheric data.

To quantify this, the ozone profiles are analysed relative to different tropopause definitions. Data is first binned in steps of  $\Delta z$ =0.1 km relative to the respective tropopause in a  $\pm 2$  km window around the tropopause. A hyperbolic tangent function is fitted to the mean values of each bin. From the fit, both the curvature and gradient at and around the tropopause can be evaluated. An example fit is shown in Appendix D1 and all fitted curves with corresponding  $R^2$  values are shown by season and tropopause definition in Appendix D2. For more meaningful comparison, the curvature is normalised to the mean ozone mixing ratio at the respective tropopause. The parameters used for normalisation are listed in Appendix C.

Analysing the behaviour of the curvature around the tropopause provides insight into the spatial relationship between tropopause and the strongest transition in ozone. Ideally, a well-placed tropopause would show the strongest curvature at the tropopause, indicating a good vertical matching to the strongest transport barrier. In the upper panels of Fig. 7, the seasonal ozone curvatures are displayed relative to their respective tropopause. The vertical positioning, shape and amplitude of the curvature profiles reveals how well the transition between troposphere and stratosphere is represented by each tropopause.

A key observation is that the highest curvature values for the dynamic<sub>1.5</sub> and dynamic<sub>2.0</sub> definitions consistently appear *above* the respective tropopause ( $\Delta z$ =0), with average displacements of 460 m and 280 m, respectively. In winter, these displacements reach up to 750 m and 340 m. These findings align with the differences in tropopause height for different PV-thresholds outlined in Appendix E. For the CARIBIC flask measurements, the 1.5 PVU and 2.0 PVU thresholds result in average tropopause heights that are 1.0 km and 0.7 km lower than the 3.5 PVU threshold. The resulting weaker curvature and gradient *at* at the tropopause for these definitions suggest a systemic underestimation of tropopause height. For all other definitions the curvature peak coincides closely with the tropopause, indicating a better alignment with the transport barrier.

Figure 6. Seasonal vertical profiles of CARIBIC  $O_3$  mixing ratios showing the distance relative to the tropopause for six definitions. The standard deviation of each bin is indicated by the error bars. The coordinates for panels (a) and (c - f) are geopotential height, while panel (b) shows the difference between the corresponding  $N_2O$  measurement and the statistically evaluated baseline value. Extra panels show the region of  $\pm 1$  km around the tropopause and ozone mixing ratios of 0–400 ppb.

Strong seasonal differences in the tropopause curvature values are observed. The curvature of the thermal tropopause is much stronger in summer than other definitions, while the chemical $O_3$  tropopause results in the highest curvature at the tropopause in winter. The overall high curvatures in summer are connected to the seasonality of ozone in the northern hemisphere: Increased downwelling in spring and higher tropopause heights in summer lead to increased  $O_3$  mixing ratios in the troposphere and near the tropopause. In the stratosphere, the  $O_3$  enhancement decreases throughout summer, until autumnal conditions are reached. The strong curvature is then caused by a combination of the downward propagation of this  $O_3$  enhancement throughout summer, while autumnal conditions are approached at higher distances above the tropopause. This seasonality is indicated in the fitted profiles shown in Fig. D2 of the appendix.






In summer, strong curvatures can be seen for most tropopause definitions, with the chemical $_{O_3}$  definition as the notable exception. Unlike other seasons, in spring a reversal of the curvature ranking can be observed: The chemical $_{O_3}$  tropopause exhibits the sharpest transition, while the thermal tropopause results in the lowest curvature. Even though vertical displacement causes the dynamic $_{1.5}$  and dynamic $_{2.0}$  tropopauses to show weak curvatures at the tropopause in winter, the total curvature amplitude remains comparable to other definitions. This suggests that the transport barrier is still being effectively represented by these dynamic definitions, although the peak transition is vertically displaced from the defined tropopause level.

The seasonal variation of dynamic tropopauses shows similarity with previous analyses. Kunz et al. (2011) and later Turhal et al. (2024) found higher PV values at the tropopause in summer and autumn for their dynamic PV-gradient tropopause. This corresponds with our findings that that the curvature at the tropopause is stronger for higher PV thresholds in these seasons. This trend inverts in spring, where both the overall amplitude and the curvature at the tropopause increases with decreasing PV-threshold. While the specifics are not clearly defined, we can therefore confirm seasonal variation of ideal PV-based tropopause definitions.

Among the definitions compared in this section, the thermal, chemical<sub>O3</sub> and dynamic<sub>3.5</sub> tropopauses show the most consistent alignment between peak curvature and tropopause level, indicating a more accurate representation of the transport barrier. However, the total and relative curvature strength strongly depends on the season. The chemical<sub>O3</sub> tropopause shows the strongest results in spring. The enhanced downward mass flux of ozone-rich air is embedded in the chemical approach, which is more relevant in this season (Holton et al., 1995). Conversely, the thin tropopause-inversion layer and strong static stability in summer enhance the thermal gradient (Gettelman and Wang, 2015). This consequently leads to the sharp transition in ozone seen in the thermal tropopause. These seasonal behaviours suggest that no single tropopause definition performs "optimally" across all conditions. Each definition responds differently to the interplay of radiative forcing, chemical transport and dynamic structure throughout the year.

Figure 7. (a–d) Climatological normalised vertical profile curvature of the ozone around the tropopause and e) the corresponding curvature at the tropopause. A hyperbolic tangent fit was applied to the region  $\pm 2\,\mathrm{km}$  around the tropopause to the tropopause-relative ozone profile for each definition. The values were then normalised to the average seasonal tropopause value of ozone. Large curvatures represents a sharp transition across the tropopause.

### 4.4 Case studies - features in small data sets




To verify the conclusions drawn from the climatological analysis in the last sections, we now investigate individual flights with high resolution  $N_2O$  and  $O_3$  data from the HALO campaign PHILEAS. We focus on the two transfer flights between Germany and Alaska on 21 August and 22–23 September 2023, which were chosen due to the large longitudinal range coverage and because they did not set a special focus on exploring atmospheric features. The first case study flight happened throughout the day on 21 August, including a stop in Reykjavík. While most of the flight takes place at pressure levels below 200 hPa and thus mostly in the stratosphere, tropopause crossings happen for all definitions during the ascents and descents. The second case study flight starts in the evening of 22 September until the morning of 23 September. The aircraft stayed well above all modelled tropopauses during the flight, except for ascent and descent. The same processing as before is applied to the measurement data, with ERA5 reanalysis parameters being interpolated onto the flight path and the statistical baseline filter being applied to  $N_2O$  observations.

Figure 8 shows data from the transfer between Germany to Alaska. The overall shape of the correlation of ozone observation to absolute PV is positive, however there is a distinct feature with low potential temperature and increasing ozone mixing ratio with V. The sorting of tropospheric and stratospheric data within this feature is significantly different for different tropopause definitions: While the chemical and dynamic tropopauses agree on the stratospheric sorting of the measurements with very high ozone (>300 ppbv) and high PV (>5 PVU), this feature does not appear to be resolved in the thermal tropopause. Using the dynamic<sub>1.5</sub> and dynamic<sub>2.0</sub> tropopauses gives a stratospheric identification for all of these high ozone mixing ratio values, while the dynamic<sub>3.5</sub> tropopause leads to large parts of this feature to be sorted into the troposphere due to a higher PV-threshold.

In the next example shown in Fig. 9, we look at the transfer flight from Alaska to Germany. Here, we see the same overall correlation between high PV values and high ozone observations. While all tropopause definitions approximately agree that the majority of measurements were taken in the stratosphere, a collection of points with high potential temperature, ozone mixing ratio of about 200 ppbv and high PV stands out. These measurements were identified as tropospheric by the chemical $_{\rm N_2O}$  tropopause but no other definition. This may be caused by the overall very low number of tropospheric points in this data, which hinders the identification of a meaningful tropospheric baseline.

# 500 5 Conclusions

Using the IAGOS-CARIBIC data set and selected flights from the PHILEAS campaign onboard HALO, we compared six commonly used tropopause definitions: The thermal tropopause based on the temperature lapse rate, dynamic tropopauses based on the  $1.5 \, \text{PVU}$ ,  $2.0 \, \text{PVU}$  and  $3.5 \, \text{PVU}$  thresholds, and chemical tropopauses based on either a climatology of  $O_3$  or a statistical baseline approach using  $N_2O$ . We found that overall the number of measurement points identified as tropospheric varies little

**Figure 8.** Ozone measurements taken during PHILEAS on 21 August 2023 from Munich to Anchorage over potential temperature. Data points are sorted into troposphere and stratosphere using different tropopause definitions, which is indicated by the colour.

between chemical $_{O_3}$ , dynamic $_{3.5}$  and the thermal tropopause, however the chemical $_{N_2O}$  shows a much stronger seasonality. Dynamic tropopauses with a PV-thresholds of 1.5 PVU or 2.0 PVU are very effective at filtering out stratospheric data, with the potential disadvantage of filtering out some tropospheric data points close to the tropopause too.

Characterising the variability of ozone in the troposphere showed that the thermal tropopause does not effectively filter out stratospheric data from the CARIBIC data set. The lower number of tropospheric points in the low PV tropopauses resulted in overall reduced variability. To quantify the sharpness of the transition from the tropospheric to the stratospheric part of the ozone profile in tropopause-relative coordinates, we calculated the curvature of the binned and fitted profiles. We found that the curvature maxima for the dynamic<sub>1.5</sub> and dynamic<sub>2.0</sub> tropopauses appeared several hundred metres above the tropopause in all seasons, indicating a vertical misalignment of the transport barrier representation. The thermal, dynamic<sub>3.5</sub> and chemical<sub>O3</sub>

**Figure 9.** As Fig. 8 for PHILEAS on 22 September 2023 from Anchorage to Munich. A number of points identified as tropospheric by the chemical  $N_{20}$  tropopause is highlighted in the top right panel.

definitions consistently align the maximum curvature with the tropopause, however the seasonal behaviour is complex. The chemical $O_3$  tropopause results in the highest curvatures in spring and winter. This may be caused by better separation of ozonerich and -poor air masses during times of stronger downward mass flux. On the other hand, the strong curvature at the thermal tropopause in summer may be related to the sharper tropopause inversion layer and stronger temperature gradients.

In analysing smaller data sets, we found benefits of using chemical tropopauses based on in-situ data over the modelled tropopauses. Especially in the case of a small-scale feature with high ozone and low potential temperature, the thermal and dynamic<sub>3.5</sub> tropopause identified points as tropospheric contrary to all other definitions. However, difficulties in applying the chemical $N_{2O}$  definitions were shown in another example, where the low fraction of tropospheric observations lead to a feature of very high potential temperature and elevated  $O_3$  mixing ratio to be identified as tropospheric. Care must therefore be taken


in choosing suitable data sets and sensible input parameters.







To conclude, we find that in the extra-tropics, the commonly used threshold values of  $1.5\,\mathrm{PVU}$  and  $2.0\,\mathrm{PVU}$  consistently underestimate the tropopause height, placing it below the strongest curvature in the ozone profile. Overall, we find the dynamic<sub>3.5</sub> definition to best represent the transport barrier of the tropopause without introducing additional variability to the tropospheric part of the data set. However, small features may be missed due to coarse resolution of reanalysis products. Chemical tropopauses were shown to work effectively in reducing tropospheric variability. The curvature at the tropopause was sharpest for the chemical<sub>03</sub> definition in spring and winter, however overall the curvature was weaker than other tropopauses. As our analysis of the chemical<sub>N2O</sub> definition was hindered by the lack of comparable vertical coordinate, we recommend further investigation in formulating a similar framework as for the chemical<sub>03</sub> tropopause. We also suggest future implementations of climatology-based chemical tropopauses to be presented using potential temperature distances to the dynamic tropopause, instead of height relative to the thermal tropopause. For such implementations, we recommend that climatologies be evaluated for several latitude bands. We further highlight the importance of high-resolution  $O_3$  and  $N_2O$  data onboard aircraft campaigns as well as global coverage of these substances at ground-level.

Data availability. The whole-air-sampler data of the IAGOS-CARIBIC project are available at https://doi.org/10.5281/zenodo.8188548, the merged (MS) high resolution IAGOS-CARIBIC data at https://doi.org/10.5281/zenodo.8188548. The observational data of the HALO flights during the PHILEAS campaign are available upon request via the HALO database (https://halo-db.pa.op.dlr.de/mission/138). The interpolated ERA5 reanalysis data for CARIBIC-2 and HALO campaign data are available at https://doi.org/10.5281/zenodo.15076519.

Author contributions. TS operated and provided data of the CARIBIC air sampler; AZ and FO operated and provided data of the CARIBIC-O<sub>3</sub> and FAIRO instruments; NE, PH, LO, and FW operated and provided data of the UMAQS instrument; and HCL contributed the ERA5 model data interpolated to the flight track. SB, TS and AE initiated and conceptualised the core research goals and together with HB and PH contributed valuable ideas in discussions. SB performed the analyses and prepared the manuscript with contributions from TS and AE. All authors contributed via discussion and comments.

Competing interests. At least one of the (co-)authors is a member of the editorial board of Atmospheric Chemistry and Physics.

Acknowledgements. This research has been supported by the Deutsche Forschungsgemeinschaft (grant no. SCHU 3258/3-1 – project ID 501095243) and DFG collaborative research programme "The Tropopause Region in a Changing Atmosphere" TRR 301 – project ID 428312742). This research was supported by the German Research Foundation (DFG) within the Priority Program HALO SPP 1294 under project number 316646266 and EN 367/19-1. IAGOS-CARIBIC data were created with support from the European Commission, na-

| ional agencies in Germany (BMBF), France (MESR) and the UK (NERC), and the IAGOS member institutions (h | ittps://www.iagos.org/ |
|---------------------------------------------------------------------------------------------------------|------------------------|
| organisation/members).                                                                                  |                        |
|                                                                                                         |                        |
|                                                                                                         |                        |
|                                                                                                         |                        |
|                                                                                                         |                        |
|                                                                                                         |                        |
|                                                                                                         |                        |
|                                                                                                         |                        |
|                                                                                                         |                        |
|                                                                                                         |                        |
|                                                                                                         |                        |
|                                                                                                         |                        |
|                                                                                                         |                        |
|                                                                                                         |                        |
|                                                                                                         |                        |
|                                                                                                         |                        |
|                                                                                                         |                        |
|                                                                                                         |                        |
|                                                                                                         |                        |
|                                                                                                         |                        |
|                                                                                                         |                        |
|                                                                                                         |                        |
|                                                                                                         |                        |
|                                                                                                         |                        |
|                                                                                                         |                        |
|                                                                                                         |                        |
|                                                                                                         |                        |
|                                                                                                         |                        |
|                                                                                                         |                        |
|                                                                                                         |                        |
|                                                                                                         |                        |
|                                                                                                         |                        |
|                                                                                                         |                        |
|                                                                                                         |                        |
|                                                                                                         |                        |

Figure A1. Mean seasonal vertical distribution of  $N_2O$  and  $O_3$ . The upper panels show the seasonal average profile over altitude as well as the mean thermal tropopause for each season. The lower panels show the distribution in height relative to the thermal tropopause. Mean values are only calculated in bins with at least 5 data points.

**Figure B1.** Distribution of tropospheric and stratospheric variability of binned CARIBIC ozone data. Measurements are separated using different tropopause definitions and binned onto a grid of latitude (1° N) and geopotential height (0.5 km). The statistical parameters describing the distributions are the mean  $(\mu)$  [ppb], standard deviation  $(\sigma)$  [ppb] and skewness  $(\gamma)$  [-].

# **Appendix C: Curvature normalisation parameters**

**Table C1.** Near-tropopause parameters for ozone in CARIBIC data. A tanh(x) fit is evaluated for the region  $\pm 2$  km around the tropopause to analyse the curvature of the vertical profile. The columns are as follows: Av. at TP - Average  $O_3$  mixing ratio at the tropopause; Curv. at TP - Absolute curvature at the tropopause; Norm. Curv. - Normalised curvature at the tropopause; Max. Curv. - Maximum (positive) curvature in the fitted region;  $\Delta z_{TP}$  (max) - Height distance between the maximum curvature and the tropopause.

| Tropopause definition  | Season | Av. at TP | Curv. at TP          | Norm. Curv.          | Max. Curv.           | $\Delta \mathbf{z}_{\mathrm{TP}}$ (max) |
|------------------------|--------|-----------|----------------------|----------------------|----------------------|-----------------------------------------|
|                        |        | [ppb]     | $[{\rm ppbkm}^{-2}]$ | $[\mathbf{km}^{-2}]$ | $[\mathbf{km}^{-2}]$ | [m]                                     |
| Chemical <sub>O3</sub> | Spring | 155.3     | 227.1                | 1.46                 | 1.47                 | 60.6                                    |
|                        | Summer | 159.1     | 140.9                | 0.89                 | 0.90                 | -101.0                                  |
|                        | Autumn | 104.9     | 75.7                 | 0.72                 | 0.72                 | 20.2                                    |
|                        | Winter | 106.7     | 179.6                | 1.68                 | 1.77                 | 141.4                                   |
| Dynamic <sub>1.5</sub> | Spring | 101.1     | 145.8                | 1.44                 | 1.80                 | 303.0                                   |
|                        | Summer | 101.2     | 153.5                | 1.52                 | 2.26                 | 343.4                                   |
|                        | Autumn | 70.5      | 45.0                 | 0.64                 | 0.82                 | 465.6                                   |
|                        | Winter | 64.8      | 50.2                 | 0.78                 | 1.27                 | 747.5                                   |
| Dynamic <sub>2.0</sub> | Spring | 121.3     | 154.7                | 1.28                 | 1.38                 | 181.8                                   |
|                        | Summer | 115.7     | 207.4                | 1.79                 | 2.24                 | 262.6                                   |
|                        | Autumn | 78.2      | 63.9                 | 0.82                 | 0.96                 | 343.4                                   |
|                        | Winter | 72.5      | 88.7                 | 1.22                 | 1.52                 | 343.4                                   |
| Dynamic <sub>3.5</sub> | Spring | 177.7     | 175.7                | 0.99                 | 1.01                 | -101.0                                  |
|                        | Summer | 159.2     | 302.2                | 1.90                 | 1.90                 | 20.2                                    |
|                        | Autumn | 104.5     | 117.3                | 1.12                 | 1.12                 | 20.2                                    |
|                        | Winter | 112.1     | 144.0                | 1.29                 | 1.33                 | -101.0                                  |
| Thermal                | Spring | 174.6     | 160.8                | 0.92                 | 0.92                 | -20.2                                   |
|                        | Summer | 155.3     | 388.3                | 2.50                 | 2.60                 | 101.0                                   |
|                        | Autumn | 103.6     | 133.3                | 1.29                 | 1.30                 | 60.6                                    |
|                        | Winter | 105.3     | 165.4                | 1.57                 | 1.57                 | 20.2                                    |

Figure D1. Binned CARIBIC  $O_3$  data on  $\Delta z$ =0.1 km and corresponding hyperbolic tangent fit. Using this fit, the gradient and curvature of the ozone profile relative to specific tropopause definitions are calculated. The  $R^2$  value gives a measure of the goodness of fit. The choice of fit function was made after visual inspection of the distribution and comparison of the goodness of fit of multiple functions, however there is no specific physical foundation for this choice.

Figure D2. The best fit of tanh(x) onto the ozone vertical profile of CARIBIC-2 data is shown for six tropopause definitions in the region of  $\pm 2$  km ( $\pm 5$  ppb for panel (b)) around the tropopause. The data was first binned in steps of 0.1 km (0.5 ppb), the mean of each bin was evaluated and the function fitted onto the resulting seasonal profiles. For each season and fitted curve, the goodness of fit is given as its  $R^2$  value.

# Appendix E: Mean tropopause height

**Table E1.** Mean height of the tropopause for the CARIBIC data set where geopotential height information is available. All values are given in kilometres [km].

| Tropopause definition  | Mean  | Spring | Summer | Autumn | Winter |
|------------------------|-------|--------|--------|--------|--------|
| Chemical <sub>O3</sub> | 10.00 | 9.69   | 10.19  | 10.16  | 9.98   |
| Dynamic <sub>1.5</sub> | 9.83  | 9.27   | 10.47  | 10.05  | 9.37   |
| Dynamic <sub>2.0</sub> | 10.21 | 9.65   | 10.84  | 10.42  | 9.77   |
| Dynamic <sub>3.5</sub> | 10.86 | 10.28  | 11.45  | 11.09  | 10.46  |
| Thermal                | 10.97 | 10.28  | 11.72  | 11.23  | 10.47  |

Figure E1. Latitude-binned ( $5^{\circ}$ ) mean seasonal tropopause height across the whole CARIBIC measurement period. Panels (a) and (b) through (f) are given in geopotential height, while in panel (b) the mean seasonal value of the statistically evaluated baseline of the chemical<sub>N2O</sub> tropopause is given. The grey shading indicates the standard deviation of the seasonal average.

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
