# Peer review of "The extratropical tropopause - Trace gas perspective on tropopause definition choice"

_EGUsphere, 2025_

## Referee Comment (RC2)

Review of

**The extratropical tropopause - Trace gas perspective on tropopause definition choice**

by Bauchinger et al.

The manuscript demonstrates the use of various tropopause definitions for the interpretation of aircraft measurements. The study is of interest for researchers working with in-situ or other high-resolution data in the UTLS region. The authors provide a detailed analysis of how the choice of the tropopause definition can impact our understanding of tracer gradients and air mass origin. These findings are further explored and illustrated with a few selected case studies. The manuscript is well written and clearly structured. I recommend publication after the following comments have been addressed.

**Major comments:**

1. The chemical ozone tropopause definition is based on a climatology that links ozone mixing ratios to a vertical distance above the thermal tropopause. The ozone profiles used to construct this climatology are from one particular station. It is not clear and not sufficiently discussed if this ozone climatology is representative for the regions of the flight campaigns. The manuscript could provide some sensitivity analysis (potentially as a supplementary material) on how climatologies based on multiple station data or satellite data would impact the presented results.

2. The motivation of the use of a chemical tropopause (given in section 3.3) argues that that the same data set can be used for the definition of the tropopause and the trace gas of interest. But this is not the case in this manuscript, or? The motivation also argues that the chemical tropopause avoids mismatches in dynamical parameters. However, with a one-station ozone climatology applied to a large geographical region it seems that these advantages of the chemical tropopause do not apply. Furthermore, it seems that the chemical tropopause is used in a different way (based on a climatology) while the dynamical tropopauses are based on the respective dynamical situation. All of this needs to be clarified and discussed more clearly.

3. To illustrate the ozone based and N2O based tropopause definitions, it might be helpful to show a latitude-altitude cross section of each gas. Or some typical profiles.

4. The use of the cross-tropopause gradient in Figure 7 seems not sufficient to assess the quality of separation into tropospheric and stratospheric air as the authors explain starting in line 374. Instead of using a supplement and whose interpretation needs to be combined with the interpretation of Figure 7, it would seem easier to evaluate the second derivative instead of the first.

5. Some of the data is not freely available as stated in the manuscript. Following the link to the HALO database, MISSION: PHILEAS webpage, I found 'Begin free data access: 2028-09-29'.

**Minor comments**

Section 2 Data: What is the time period considered? How many flights are included in the analysis?

Line 83: in-fight -> in flight

Line 21 (significant uncertainties in our understanding) and line 25 (knowledge gaps and uncertainties): I do not disagree, but these general statements need to be explained and backed up more. What are the uncertainties and remaining questions? Describe the general knowledge gaps and give some examples illustrating why these are relevant.

Line 98: I don't really understand the argument: '*the routes taken were not tailored to explore special features in the atmosphere*'.

Line 110: Was 'MS data set' defines?

Line 250: This seems not really true north of 40N.

Line 258-263: Can downwelling impact tracer distributions and PV in different ways and therefore impact the differences in seasonality?

Line 269-270: The statement 'most notable … spring' is not clear to me.

Line 286: Does 'with' need to be removed. Something seems wrong in this sentence.

Line 286-291: Wouldn't this impact ozone and thus the chemical ozone tropopause in similar ways?

Line 295 and other places: The term seasonal trends seems to refer to seasonal variations and might be confusing.

Line 308-309: Does this show in the mean or in the standard deviation?

Figure 6: For comparability, it would be better to show the full y-range for the upper panels as well.

---

## Author Comment (AC1)

**Response to Reviewer 1**

We thank the reviewer for thorough reading of our manuscript and providing valuable feedback. The comments and remarks have aided in improving the analysis and manuscript. We understand the main concerns are that (i) the impact of STE on tropopauses and $O_3$ variability as tropopause-evaluation metric is not sufficiently addressed, (ii) the cross-tropopause gradient is not suitable as stand-alone assessment tool of tropopause-sharpness and (iii) the context of the case study flights is not discussed in enough detail.

In the following, we address all comments in detail (Reviewer's comments in italic, quotations of the corresponding revised text passages in blue).

Based on suggestions from all reviewers, we have added a new Section 3.4 titled "Key differences in tropopause definition methodologies". Additionally, the quantification of tropopause-sharpness has been changed from a calculation of the cross-tropopause gradient to the curvature at the tropopause.

**General Comments**

**(i) Impact of STE on $O_3$ variability as a metric**

*On utilizing ozone variability as an assessment tool for reliable discrimination of troposphere and stratosphere observations (i.e., Section 4.2): I'm not convinced this approach is a reliable assessment of tropopause definition performance. One particularly problematic condition to consider (which is not discussed at length in the present draft) is stratosphere-troposphere exchange (STE). Recent STE will undoubtedly result in air that chemically appears to be stratospheric/tropospheric but is located in the troposphere/stratosphere (respectively). Thus, using variability as an indicator of performance/appropriateness would seem to reward definitions that "remove" or otherwise minimize the inclusion of STE events. This complexity should at the very least be acknowledged and considered further in the interpretation of the meaning/significance of these results.*

We agree that a more thorough discussion of the differences between tropopause definitions that are due to different calculation approaches should have been presented more clearly. Generally speaking, this paper aims to highlight the (dis-)advantages and sensitivities of the different definitions without referencing specific meteorological situations. We compare three main types of tropopauses, which are being used in literature for varieties of studies. While the chemical tropopauses are more representative of an air mass' origin and thus represent recent STE / tropopause folds with an emphasis on what the current chemical composition is representative of, other tropopause definitions put the focus on the location within large structures (aka representing the tropopause through the means of global almost-continuous surfaces).

We chose to use the variability of ozone relative to the tropopause as a measure of how well the natural variability of ozone can be disentangled from dynamic processes in the atmosphere. As discussed e.g. in Millán et al. (2023), separating these effects from each other is necessary to calculate for example long-term trends in ozone. Keeping this in mind, we cannot say that one single tropopause definition will be better for all types of studies. We merely aim to increase the knowledge base of benefits and limitations of the tropopauses studies in the context of aiming to homogenise air masses over long

spatial and temporal scales.

We have added some discussion of STE in the newly added Sect. 3.4 Key differences in tropopause definition methodologies.

(ii) **Cross-tropopause gradient**
*On the cross-tropopause gradient (i.e., Figure 7): using the magnitude of the ozone gradient across the tropopause as an assessment tool also seems problematic to me. One could achieve the largest gradient by simply introducing a high bias in tropopause altitude such that the ubiquitous sharp stratospheric increase in ozone contributes entirely to the diagnosed cross-tropopause gradient. I think the alternative approach of comparing (or rather, differencing) the gradients/slopes in ozone vs. altitude BELOW and ABOVE tropopause would be a more meaningful and reliable assessment of the appropriateness of each definition. That is nearly accomplished here, but this slight adjustment in approach would resolve an otherwise misleading means of assessing performance.*

Taking into account comments from both reviewers, we have revised this part of the analysis to focus on the curvature of the vertical profile at the tropopause to better highlight the "sharpness" of the transition. However, we would like to clarify the use of the normalised tropopause gradient in the analysis: 1) The high bias in the gradient for tropopauses at a higher average ozone value was taken into account through the normalisation to the average tropopause value. As could be seen in the previous Appendix B1, the tropopause averages varied from 67-160 ppb. For tropopause definitions with a high bias, this normalisation factor also increases, which in turn reduces the normalised gradient. 2) Additionally, while admittedly not the focus of the analysis, we make reference to the difference between the gradients at the tropopause as well as 0.5 km below in the text. We agree that the non-normalised cross-tropopause gradient should not be used as a stand-alone assessment tool.

We address this comment through the revision of Sect. 4.3 (formerly "Vertical profiles and cross-tropopause gradient", now "Vertical profiles and curvature at the tropopause"). The gradient evaluation has been replaced by an assessment of the curvature at the tropopause, which we believe to be a stronger metric for evaluating the transition happening at the tropopause. For this, we describe the region $\pm 2$ km around the tropopause with a fit and evaluate the curvature of this fit for each season. A larger curvature then represents a sharper transition between the characteristic tropospheric and stratospheric parts of the vertical profile of ozone.

(iii) **Case study** *On the case study analyses in Section 4.4: this analysis was very brief and I found to be minimally convincing. In particular, where are the observations contextually, especially those in Figure 8? Is the high ozone portion of the flight flagged as tropospheric in panels (e) and (f) of Figure 8 the result of an STE event? For example, could this be a tropopause fold? To convincingly demonstrate appropriateness of these definitions for a case study, more information should be given. Though tracer-tracer analysis was mentioned early, I thought its exclusion from this study was a missed opportunity.*

Thank you, this comment poses some very interesting questions. We would like to again highlight that the focus of this paper does not lie in specifically exploring features of the atmosphere, but in showing differences between tropopause definitions. In order to briefly showcase the potential impact,

we present case studies of generally non-specific meteorological scenarios. The benefits and limitations of in-situ calculable versus modelled tropopause parameters is visualised through these flights and later discussed in more detail.

For more context, you may refer to Fig. 1 of this response showing curtains of potential vorticity (PV), temperature and potential temperature from ERA5 reanalysis data combined with the interpolated modelled tropopause pressures. In the left panel, the stop-over in Reykjavík commencing around 12:00 corresponds to a sharp descent and corresponding crossing of the tropopause. Due to changes in PV, temperature and potential temperature, the modelled dynamic and thermal tropopause show some unusual features in this region, leading to the differences between definitions described. We have added the following information in Lines 474–478 to provide the reader with more context on the case studies:

The first case study flight happened throughout the day on 21 August, including a stop in Reykjavík. While most of the flight takes place at pressure levels below 200 hPa and thus mostly in the stratosphere, tropopause crossings happen for all definitions during the ascents and descents. The second case study flight starts in the evening of 22 September until the morning of 23 September. The aircraft stayed well above all modelled tropopauses during the flight, except for ascent and descent.

**Technical Edits**

*Line 9: "larger variability near the tropopause" would be better stated as "larger composition variability near the tropopause"*
Thank you, the wording has been changed as suggested in Line 9.

*Line 107: "substances" should be "substance"*
Thank you, this has been changed in Line 111.

*Line 151: "across the tropopause" would be better stated as "from troposphere to stratosphere in the extratropics"*
We agree, the revised phrasing has been implemented in Lines 156–157.
Due to stratification above the tropopause, PV experiences a strong increase from troposphere to stratosphere in the extra-tropics, allowing for the definition of a *dynamic* tropopause.

*Line 153: "the vast majority of" could be stated simply as "most"*
Thank you for the hint, the sentence has been changed accordingly in Line 158.

*Line 179: suggest revising "last decades" to "last several decades"*
Thank you, we have decided to remove that part of the sentence in Line 185.

*Line 304: I find the statement "a large number of" to be somewhat misleading. What is considered large? Could you express this as a fraction of all aberrations? From my interpretation of the analysis, it appears to have a minimal impact on the statistics, so that gives the impression that these observations are still few though certainly more numerous than remaining definitions.*
We agree this should have been quantified better. The following sentence has been added in Lines 353–354 to expand on this point:

[Figure]

Figure 1: ERA5 reanalysis curtains interpolated onto the flight track of the case study HALO-PHILEAS flights on 21 Aug 2023 and 22-23 Sep 2023. Potential vorticity (PV), temperature (T) and potential temperature (THETA) are shown for the pressure range 500-100 hPa on a vertical resolution of ... and a temporal resolution of 30 minutes.

Note that over 4% of the points sorted into the troposphere by the thermal tropopause have ozone mixing ratios over 200 ppb, compared to 1.5% for the dynamic$_{3.5}$ tropopause and less than 0.1% for all other definitions.

*Line 313: "stratosphere" should be "stratospheric"*

Thank you, this has been changed in Line 366.

*Lines 322-323: this sentence also appears to be an overstatement. The variability in ozone is not largest for the thermal tropopause in all seasons, though the range appears to be. In particular, the mode is largest for the N2O definition in autumn and winter.*

Indeed, we have changed this sentence in Lines 376–378 to better represent the results: For all seasons, the spread of tropospheric bin variabilities of ozone is largest with the thermal tropopause with corresponding high modes in spring and summer, although in autumn and winter the chemical$_{N_2O}$ tropopause results in a higher variability mode.

*Line 338: delete floating paren after "Fig. 6b"*
Thank you, we have removed this for Line 393.

**References**

Millán, L. F., Manney, G. L., Boenisch, H., Hegglin, M. I., Hoor, P., Kunkel, D., Leblanc, T., Petropavlovskikh, I., Walker, K., Wargan, K., and Zahn, A.: Multi-Parameter Dynamical Diagnostics for Upper Tropospheric and Lower Stratospheric Studies, Atmospheric Measurement Techniques, 16, 2957–2988, https://doi.org/10.5194/amt-16-2957-2023, 2023.

---

## Author Comment (AC2)

**Response to Reviewer 2**

We thank the reviewer for thorough reading of our manuscript and providing valuable feedback. We understand the two main concerns as (i) the representativeness of the ozone climatology for CARIBIC data is not sufficiently discussed and (ii) that a more thorough discussion of methodological differences between tropopause definitions and corresponding sensitivities was needed.

In the following, we address all comments in detail (Reviewer's comments in italic, quotations of the corresponding revised text passages in blue).

Based on suggestions from all reviewers, we have added a new Section 3.4 titled "Key differences in tropopause definition methodologies". Additionally, the quantification of tropopause-sharpness has been changed from a calculation of the cross-tropopause gradient to the curvature at the tropopause, with the title of Sec. 4.3 changed to "Vertical profiles and curvature at the tropopause" to reflect this change.

**General Comments**

**(i) Representativeness of Ozone Climatology**

*The chemical ozone tropopause definition is based on a climatology that links ozone mixing ratios to a vertical distance above the thermal tropopause. The ozone profiles used to construct this climatology are from one particular station. It is not clear and not sufficiently discussed if this ozone climatology is representative for the regions of the flight campaigns. The manuscript could provide some sensitivity analysis (potentially as a supplementary material) on how climatologies based on multiple station data or satellite data would impact the presented results.*

Thank you, we agree that a more thorough discussion of the ozone climatology representativeness would certainly be interesting. The study on hand focussed on the differences between tropopause definitions as they are currently commonly implemented. We therefore chose to use the chemical ozone tropopause that is available as part of the CARIBIC data set and applied the same processing to the PHILEAS observations for consistency. The sensitivity analysis would therefore, while certainly adding an interesting component, be better placed in future studies. To clarify the studies' intent, we have added the following in Lines 53–55 to the manuscript:

To allow comparability with previous studies, we put a focus on tropopause definition parameters as available in published data sets, rather than separate implementations..

**(ii) Chemical ozone tropopause application**

*The motivation of the use of a chemical tropopause (given in section 3.3) argues that that the same data set can be used for the definition of the tropopause and the trace gas of interest. But this is not the case in this manuscript, or? The motivation also argues that the chemical tropopause avoids mismatches in dynamical parameters. However, with a one-station ozone climatology applied to a large geographical region it seems that these advantages of the chemical tropopause do not apply. Furthermore, it seems that the chemical tropopause is used in a different way (based on a climatology) while the dynamical tropopauses are based on the respective dynamical situation. All of this needs to be clarified and dis-*

*cussed more clearly.*

The argument for using chemical tropopauses refers to the approach of deriving the tropopause from one tracer and to apply this result to other observables. The reference to the same data set in this case refers to the type of measurement being in-situ measurements / air sampling, rather than trying to match this with modelled or remote-sensing-derived quantities. The ozone climatology provides the basis for calculating representative height-differences to the tropopause for the chemical ozone tropopause, however specific measurements are then used to match each observation with a representative tropopause value. This stands in contrast to the modelled thermal and dynamic definitions, which provide the general larger-scale meteorological context for each observation on a pre-defined grid. The tropopause height is then an independent variable to the observations.

We have extended the discussion of differences between tropopause definitions through the newly added Section 3.4 "Key differences in tropopause definition methodologies".

**(iii) $O_3$ and $N_2O$ typical profiles**

*To illustrate the ozone based and N2O based tropopause definitions, it might be helpful to show a latitude-altitude cross section of each gas. Or some typical profiles.*

Thank you, we have added the mean profiles of $O_3$ and $N_2O$ showcasing the typical vertical distribution of these gases as Fig. A1 in the appendix. The figure is also shown as Fig. 1 of this response.

[Figure]

Figure 1: Seasonally averaged profiles of $O_3$ and $N_2O$ in CARIBIC data. Top two panels show vertical distribution over geopotential height, while the lower to panels show the averaged distribution relative to the local thermal tropopause.

(iv) **Cross-tropopause gradient**

*The use of the cross-tropopause gradient in Figure 7 seems not sufficient to assess the quality of separation into tropospheric and stratospheric air as the authors explain starting in line 374. Instead of using a supplement and whose interpretation needs to be combined with the interpretation of Figure 7, it would seem easier to evaluate the second derivative instead of the first.*

Thank you, we agree that the second derivative provides a much stronger indication of this separation. We have revised this section and now present the curvature at the tropopause in Section 4.3. "Vertical profiles and curvature at the tropopause".

(v) **Data availability**

*Some of the data is not freely available as stated in the manuscript. Following the link to the HALO database, MISSION: PHILEAS webpage, I found 'Begin free data access: 2028-09-29'.*

We agree that this should be stated clearly. Moreover, we have since made the ERA5 interpolated data accessible online. The data availability section in Lines 536–539 now reads:

The whole-air-sampler data of the IAGOS-CARIBIC project are available at `https://doi.org/10.5281/zenodo.8188548`, the merged (MS) high resolution IAGOS-CARIBIC data at `https://doi.org/10.5281/zenodo.8188548`. The observational data of the HALO flights during the PHILEAS campaign are available upon request via the HALO database (`https://halo-db.pa.op.dlr.de/mission/138`). The interpolated ERA5 reanalysis data for CARIBIC-2 and HALO campaign data are available at `https://doi.org/10.5281/zenodo.15076519`.

**Minor comments**

*Section 2 Data: What is the time period considered? How many flights are included in the analysis?*
Thank you, we have added the following sentence in Lines 70–71:
The dataset used for this study includes 278 individual flights, taking place regularly between 2005 and 2020.

*Line 21 (significant uncertainties in our understanding) and line 25 (knowledge gaps and uncertainties): I do not disagree, but these general statements need to be explained and backed up more. What are the uncertainties and remaining questions? Describe the general knowledge gaps and give some examples illustrating why these are relevant.*
Thank you for the comment, we have revised the paragraph with additional information. Lines 20–27 now read as:
" However, in-situ observations in this complex region are rare, leading to significant uncertainties in our understanding of processes such as stratosphere-troposphere exchange (STE) or long-term trends in atmospheric circulations. Despite growing scientific interest and research in this region over the past few decades, key knowledge gaps remain: For example, while changes in the strength of the jet streams and the Brewer-Dobson circulation are expected, the extent and implications are not yet well understood (Butchart and Remsberg, 1986). Ozone trends in the exUTLS are also difficult to quantify due to the complex interplay between processes such as large-scale circulations and STE, which contributes to a large natural variability. These uncertainties extend to pollutant transport and consequently the overall impact of aerosols or short-lived trace gas emissions."

*Line 83: in-fight -> in flight*
Thank you, this has been changed in Line 88.

*Line 98: I don't really understand the argument: 'the routes taken were not tailored to explore special features in the atmosphere'.*
This line refers to the nature of the chosen flights as commercial routes, rather than scientific flights aiming to specifically sample air in "interesting" conditions. This approach stands in contrast to (scientific) flight routes aiming to "feature-hunt" and presents a more representative data set for typical states of the atmosphere. The same reason was applied when choosing the flights for the case studies: Rather than taking data from scientific flights during the PHILEAS campaign, the transfer flights were chosen to be more representative for the atmosphere. No changes have been made to the manuscript.

*Line 110: Was 'MS data set' defines?*
Thank you, we have changed the sentence to explain the type of the data instead:
The $O_3$ data from CARIBIC used here is part of the merged data set on $10\,s$ time resolution

*Line 250: This seems not really true north of 40N.*
We agree and have changed Line 292 to read:
In contrast, both chemical tropopause definitions result in lower seasonal variability at latitudes between 30-50°N.

*Line 258-263: Can downwelling impact tracer distributions and PV in different ways and therefore impact the differences in seasonality?*
Indeed, the difference in the sensitivity to downwelling of different tropopause definitions will impact the seasonal differences between them. We have added the following sentences in Lines 305–307:
The different sensitivity of tropopauses to downwelling may additionally impact the seasonal variations. Changes in the chemical composition through increased downwelling in some seasons will immediately be represented by the chemical tropopauses, changes in PV may be more complex.
*Line 269-270: The statement 'most notable ... spring' is not clear to me.* We have changed the wording to express the content more clearly in Lines 312–315:
There are some slight differences between these definitions. In winter, the dynamic$_{3.5}$ tropopause identifies a larger fraction of tropospheric air in the data set, while in spring the stratospheric fraction is higher for the chemical$_{O_3}$ tropopause than the other two. However, the overall seasonal variations are very similar for these definitions.

*Line 286: Does 'with' need to be removed. Something seems wrong in this sentence.* We have improved the readability of the sentence in Lines 330–331:
While the chemical$_{O_3}$ tropopause results in a similar tropospheric fraction as the thermal and dynamic$_{3.5}$ tropopauses, the tropospheric fraction is much higher in autumn for the chemical$_{N_2O}$ tropopause.

*Line 286-291: Wouldn't this impact ozone and thus the chemical ozone tropopause in similar ways?*
Thank you, we agree that this is not clearly explained in the text. We have added the following clarification in Lines 333–340:
This increased tropospheric fraction may be caused by the sensitivity of the statistical baseline algorithm on the underlying dataset. The weakening of the subtropical jet and corresponding flushing of tropical air into the extra-tropics results in higher $N_2O$ mixing ratios in the UTLS. The baseline algorithm may then identify the mixed air masses as tropospheric, while other definitions are less sensitive to these chemical composition changes. This effect appears to then not be apparent for the chemical$_{O_3}$ tropopause, as the climatology at the tropopause already includes this seasonality. Another effect causing this difference could be the seasonal variation of $N_2O$ emissions propagating upwards. A systematic investigation of the statistical baseline algorithm across different seasons and conditions is recommended to better understand this behaviour.

*Line 295 and other places: The term seasonal trends seems to refer to seasonal variations and might be confusing.*
Thank you, we have changed this in Lines 315, 344 and 399.

*Line 308-309: Does this show in the mean or in the standard deviation?*
Thank you, this is an important point. We have added the following information in Lines 359–362:
For these definitions, 16% and 12% of stratospheric ozone mixing ratios were below 100 ppb, compared to 6% or less for other tropopause definitions. This results in strong decreases of the mean mixing ratio across all seasons and increases in the variability in regions close to the tropopause. Overall, the mean ozone value is shown to strongly depend on the tropopause definition, leading to a range of 60 ppb in the stratosphere and 21 ppb in the troposphere.

*Figure 6: For comparability, it would be better to show the full y-range for the upper panels as well.*
We have adjusted Fig. 6 accordingly.

**References**

Butchart, N. and Remsberg, E. E.: The Area of the Stratospheric Polar Vortex as a Diagnostic for Tracer Transport on an Isentropic Surface, Journal of the Atmospheric Sciences, 43, 1319–1339, https://doi.org/10.1175/1520-0469(1986)043<1319:TAOTSP>2.0.CO;2, 1986.

---

## Referee Report (RR1)

**Second review of**

**The extratropical tropopause - Trace gas perspective on tropopause definition choice**

by Bauchinger et al.

Overall, the authors have adequately responded to my concerns given in the first review. In the revised version of the manuscript, the authors have 1) added a section on "Key differences in tropopause definition methodologies", 2) added a figure of the typical vertical distribution of O3 and N2O and 3) added a section on "Vertical profiles and curvature at the tropopause".

I can understand that the authors don't want to add a complex sensitivity study on the impact of the one-station ozone climatology on the chemical tropopause results. I agree that it is acceptable to leave this point for future discussions. However, it would be good to communicate this point with the future audience of the paper. I would suggest that the authors add 1-2 sentences on the fact that the representatives and potential impact of the one-station ozone climatology are not explored here and subject of future work.

---

## Author Response (AR2)

**Response to the Editor**

The authors thank the editor and reviewers for providing detailed and valuable feedback. We have addressed the remaining comments in the manuscript.

**Response to Reviewer 1**

Line 249: "relative" should be "relative to the tropopause" Thank you, we have changed this in Line 250.

Line 446: for which season?

We have clarified the paragraph by changing "For this season" to "In summer" in Line 448.

Lines 453-454: parenthetical citations should be in-text

Thank you, we have adjusted the citation for Lines 453-454.

Figure 7: the colors here are difficult to distinguish, especially the blues and oranges. Further, the mix of oranges, red, and green is not readily discernible for those with color-vision deficiency. Lastly, I don't see any of the light blue lines (i.e., N2O)...

Thank you, we agree that the colours are not as easily distinguishable as they should be. The  $N_2O$  label also should not have appeared in the legend. We have decided to introduce different line-styles to allow differentiation, which we also show as Fig. 1 in this response.

**Response to Reviewer 2**

I can understand that the authors don't want to add a complex sensitivity study on the impact of the one-station ozone climatology on the chemical tropopause results. I agree that it is acceptable to leave this point for future discussions. However, it would be good to communicate this point with the future audience of the paper. I would suggest that the authors add 1-2 sentences on the fact that the representatives and potential impact of the one-station ozone climatology are not explored here and subject of future work.

Thank you, we agree that this point needs to be discussed in the manuscript. We have added the following sentences in Lines 194–195:

It should additionally be noted that the representativeness of this single-station ozone climatology is not explored in detail here. The sensitivity to latitudinal differences in the ozone distribution may be of interest for future work.

Figure 1: Updated Figure 7 from the manuscript.